# Influencing Factors and Group Differences of Urban Consumers’ Willingness to Pay for Low-Carbon Agricultural Products in China

**DOI:** 10.3390/ijerph20010358

**Published:** 2022-12-26

**Authors:** Ning Geng, Zengjin Liu, Xibing Han, Xiaoyu Zhang

**Affiliations:** 1School of Public Administration, Shandong Normal University, Jinan 250014, China; 2Shanghai Academy of Agricultural Sciences, Shanghai 201403, China

**Keywords:** low-carbon agricultural products, willingness to pay, influencing factors, group differences

## Abstract

Developing low-carbon agriculture has become a development goal for low-carbon economies in various countries, and consumers’ awareness and willingness to pay (WTP) for low-carbon agricultural products is an important link in achieving the sustainable development of low-carbon agriculture. The theory of planned behavior is a widely used framework to explain consumers’ food choices. Considering the intrinsic norms of consumers, their perceptions of low-carbon agricultural products, and shifts in consumer behavior, our study adds the influence of environmental awareness and consumer preferences to the theoretical framework of analysis. We choose the contingent valuing method (CVM) and use 532 consumer questionnaires in Shanghai to validate Chinese urban consumers’ WTP for low-carbon products and its influencing factors. The findings show that Chinese urban consumers have a high overall awareness of low-carbon agricultural products and, after strengthening the conceptual information of consumers, most consumers agree that low-carbon vegetables are more conducive to ecological environment protection, quality, and safety guarantees than conventional vegetables. The existing analysis showed that some variables such as bid price, behavioral attitudes, subjective norms, and consumption preferences significantly influenced consumers’ willingness to pay for low-carbon leafy greens, while the effect of the environmental awareness variable was not significant. Further research found that consumers’ WTP for low-carbon leafy greens showed significant group differences across income, gender, age, and education. Therefore, to promote the consumption of low-carbon agricultural products in China, we should attach importance to the publicity and guidance of low-carbon vegetables and strengthen the certification of low-carbon vegetable products. This study can provide policy reference for reasonably regulating and subdividing China’s low-carbon agricultural products market.

## 1. Introduction

With global warming and its resulting ecological environment, industrial and agricultural production, social, economic, and human health issues, the development of low energy consumption, low emissions, and low pollution of the economic model has become the goal of low-carbon economic development of all countries [1]. As an important part of the economic sector, the development of low-carbon agriculture has far-reaching significance for a country’s ecological environment and socio-economic development [2]. Since the reform and opening up, China’s agriculture has developed from traditional farming to mechanized and technologically advanced modern agricultural production. The extensive use of chemical fertilizers and pesticides has increased the yield, but also caused problems such as decreased soil fertility, excessive toxic substances, and excessive emission of greenhouse gases [1]. Therefore, promoting low-carbon agriculture can not only reduce carbon emissions but also improve soil quality, optimize the agricultural production environment, and truly achieve sustainable agricultural development. However, consumers, as the ultimate demanders of low-carbon agricultural products, determine whether the market for low-carbon agricultural products can exist and develop in the long term. Research on consumers’ perceptions of low-carbon agricultural products, their willingness to choose consumption methods of low-carbon agricultural products, and their willingness to pay is important for promoting the establishment of low-carbon consumption patterns, optimizing low-carbon agricultural market policies, and achieving low-carbon economic goals [3,4].

Since the 1990s, low-carbon food has become one of the ecological labels of China’s food production industry [5]. Low-carbon agricultural products mainly refer to agricultural products with low energy consumption and low emissions in line with the product life cycle by combining the low-carbon concept with industrial development on the basis of conforming to pollution-free and green production standards [6]. There are currently two ways to identify low-carbon agricultural products for the consumer market. One is the certification of low-carbon agricultural products, which guides the public’s consumption choices by awarding low-carbon marks to products and encouraging enterprises to develop low-carbon product technologies, ultimately leading to a reduction in global greenhouse gases [5,6,7]. The second is the use of a carbon label, where the greenhouse gases emitted by agricultural products throughout their life cycle (carbon footprint) are indicated on the product label with a quantitative index, informing consumers of the product’s carbon information in the form of a label [8]. Research has already focused on the measurement of product carbon emissions and the implementation of product carbon labeling. The UK first introduced carbon labeling in 2007, followed by France, the US, South Korea, and other developed countries that have attempted to institutionalize carbon footprint certification by putting product carbon footprints on labels [8]. Taiwan announced carbon labeling and implemented its use in 2009, and Japan officially implemented a carbon labeling system for agricultural products in 2011 to communicate to consumers the carbon dioxide emissions from the production process of agricultural products through environmental labels. These two approaches can regulate the market for low-carbon agricultural products more effectively [9], promote the supply of low-carbon information, and thus alleviate the problem of low-carbon information asymmetry [10].

More relevant research on low-carbon agriculture is currently focusing on carbon emissions from agricultural production at the national or industry level. There is a strong link between agricultural production and carbon emissions [11] and, according to the 2007 IPCC report (the Intergovernmental Panel on Climate Change), agricultural greenhouse gases account for 10–20% of greenhouse gases generated by human activities. Agricultural greenhouse gas emissions contribute to the greenhouse effect while seriously affecting ecological stability [12]. In 2008, agricultural emissions in the United States were 4.28108 tons of carbon, of which farming was 1.108 times greater than livestock emissions [13]. From field studies in developing countries, such as Peru and Kenya, it appears that villages and ecosystems are affected by agricultural carbon sources and sinks [14]. In terms of soil carbon sequestration capacity, many countries do not pay enough attention to carbon emissions from agricultural operations [15]. In addition to developing policies to save energy and reduce emissions, for developing countries, a change in the way agriculture is developed is needed to achieve carbon reduction [16]. In addition, the development of low-carbon agriculture mainly considers the overall benefits, i.e., the comprehensive evaluation of economic, ecological, and social benefits. Especially for farms in the upper part of the chain, low-carbon agriculture may have benefits that are less than the costs in the short term but, in the long term, good economic and ecological benefits can be obtained [17].

Research on low-carbon agricultural production may neglect the end-consumer market in the chain. International research on carbon-labeled products and consumer attitudes, choices, behaviors and consumption patterns have obtained corresponding results [18,19,20,21,22]. However, there are few studies on the willingness to pay for low-carbon agricultural products. Theories of consumer behavior suggest that consumers are motivated by external stimuli and search for information about products before they develop a need and, after a series of complex psychological activities, they evaluate whether they have the ability to satisfy their own needs and finally make a purchase decision and provide feedback [23]. Atsushi et al. (2010) used a choice consumption experiment to investigate the interaction between access to information and the value of carbon-labeled food products in consumers’ minds [24], concluding that consumers’ willingness to pay was significantly higher when they were actively seeking information than when they were passively receiving it, and the environmental information contained in low-carbon products was beneficial in creating value in consumers’ minds. Research findings show that 72% of EU residents support carbon labeling as a mandatory label and that carbon labels have a positive effect on promoting low-carbon product choice behavior [25]. Carbon labeling has received much attention globally as a way to both influence consumer choice behavior and to specifically quantify the source of carbon emissions. In terms of the factors influencing low-carbon consumption, psychological attribution [26], policy interventions [27], awareness, education, and consumer preferences [14] significantly influence Chinese consumers’ low-carbon agricultural product consumption behavior. In addition, sustainable low-carbon consumption behavior is also influenced by five main factors: behavioral intention, behavior-specific knowledge and skills, situational factors, age, and gender [22,24,28]. By constructing an analysis of low-carbon consumption preferences and demand for low-carbon products, the study found that low-carbon products have two important characteristics: a binary value structure and consumption preferences that go beyond basic values.

In the literature on consumption WTP, scholars consider the Contingent Valuing Method (CVM) to be the most appropriate method among many others. Compared to developed countries, the application of CVM in China lags behind. It is mainly used in the evaluation of environmental resources and more in the fields of air quality, ecosystem, tourism resources, and biodiversity [29]. In Germany, a study of the willingness to pay for climate-friendly products showed that the majority of the country’s citizens support climate change and renewable energy regulations, but public awareness does not influence individual consumption behavior. Market consumption surveys show that the percentage of consumers willing to pay more for climate-friendly products is 4.2% in Germany, 6.3% in Spain, 12.1% in France, 19.1% in the UK, and 22.2% in the US (Allianz, 2011) [30]. A survey of the willingness to pay for low-carbon agricultural products in Japan showed that consumers would be willing to buy rice at 24% above current prices and tomatoes at 15% above current prices if there was a 100% CO2 reduction in the production process of agricultural products [19]. Based on survey data from Chinese urban consumers, the conditional value assessment method was used to estimate that consumers are willing to accept a certain amount of price increase for low-carbon pork and are willing to accept a 10% price increase over normal pork. The factors influencing the purchase of low-carbon pork were also examined, with low-carbon pork price, consumer awareness of low-carbon products, household income, household size, and education level all having significant effects on consumers’ willingness to pay [20]. This shows that the CVM method is quite well established for assessing the willingness to pay for environmental resource values and is also suitable as a research method for this paper on consumers’ willingness to pay for low-carbon agriculture products.

Worldwide, consumers’ behavior has undergone significant changes in terms of food consumption and has been strongly influenced by various ecological environment variables. In particular, it is necessary to further study the willingness of consumers in developing countries to pay for low-carbon agricultural products and its influencing factors, which cannot be ignored in promoting the sustainable development of low-carbon agriculture in developing countries. The marginal academic contributions of this paper are (1) using the hypothetical value assessment method to design a questionnaire to empirically analyze consumers’ willingness to pay for low-carbon leafy greens and their influencing factors, taking low-carbon vegetable products as an example, which provides important methodological support for the study of low-carbon agricultural products consumption in China; (2) to identify the differences in the influence of ecological and environmental protection variables on Chinese consumers’ willingness to pay for low-carbon products and to provide a realistic basis for further segmentation of the Chinese low-carbon product market; and (3) to calculate the average consumers’ willingness to pay for low-carbon vegetables and then analyze the group differences in willingness to pay for low-carbon vegetables among different consumer groups in order to provide a reference for the formulation of low-carbon agricultural market policies in China.

## 2. Theoretical Framework and Model Construction

### 2.1. Theoretical Framework

Low-carbon agricultural products are processed and produced by the whole system chain with minimal greenhouse gas output, covering the whole process of production, processing, packaging, distribution, and consumption of agricultural products from farm to table [29]. As a kind of selective consumption, consumers’ willingness to pay for low-carbon agricultural products is a reflection of consumers’ attitudes based on their individual cognition and consumption preferences, such as the comprehensive evaluation of the price, nutrition, and trust of low-carbon agricultural products. On the one hand, individual cognition is influenced by the theory of planned behavior. Behavioral attitudes refer to the positive or negative feelings that consumers hold about selective behavior. In 1934, Lapie conducted survey research and found that there is an inconsistency between individual attitudes and actual behavior [31]. Ajzen then proposed the theory of planned behavior, which extended the theory of rational behavior by adding a third determinant of behavioral tendencies, namely perceived behavioral control, making explanations and predictions of behavioral choices more plausible [32]. Based on the theory of consumer behavior, the theory of planned behavior, the empirical findings of the existing literature, and from the consumers’ own microscopic point of view, environmental protection consciousness, behavior attitude, subjective norms, perceived behavioral control, consumer preferences, and individual and family characteristics were used to establish an analytical framework of the factors influencing consumers’ WTP for low-carbon agricultural products (Figure 1).

#### 2.1.1. Environmental Protection Consciousness

An important value of developing a low-carbon vegetable industry is that it contributes to ecological conservation, so environmentally conscious consumers are more likely to be willing to buy low-carbon leafy greens. In particular, the perception of “low-carbon products” and their quality and safety also influence consumers’ willingness to pay. Therefore, three variables, namely “environmental awareness”, “confidence in purchasing vegetables” and “low-carbon awareness”, are chosen to measure consumers’ environmental awareness.

#### 2.1.2. Behavior Attitude

Attitudes assess the extent to which people are in favor or against the topic under discussion [33]. Consumer decision-making is a complex psychological process and there is a large body of empirical research on the attitudes that influence consumers’ choice of environmentally friendly foods. The results of these studies indicate a significant positive relationship between consumer attitudes toward organic food [34,35], green food [36], sustainable products, and purchase intentions [37]. This paper attempts to determine consumer attitudes in terms of both ecological conservation, and quality and safety assurance of low-carbon vegetables.

#### 2.1.3. Subjective Norms

Subjective norms are associated with perceived social influence or pressure to engage or disengage from a particular behavior [36]. Subjective norms also reveal the influence of an individual’s behavior on a reference group [37]. The most important influences associated with consumers’ purchase of environmentally friendly food may originate from their friends, family, or government [38]. Previous findings suggest that subjective norms have a positive influence on consumers’ behavioral intentions toward food choices [39,40]. Based on the above discussion, we chose two variables to replace the factor of subjective norms: consumption trends and government calls.

#### 2.1.4. Perceived Behavioral Control

Perceptual behavioral control is the ability of individuals to control their own behavior alone [36]. The results of previous related studies suggest that perceptual behavioral control is a key factor influencing consumers’ purchases of green foods [41,42]. Since low-carbon and green products share some common attributes, the results of consumer research on green products can inform the research on low-carbon products. Another study has shown [38] that willingness to pay for organic food is influenced by the unavailability of the price factor. Therefore, price payment expectations and degree of willingness to purchase can be used as valid explanatory variables.

#### 2.1.5. Consumer Preferences

Behavioral habits are automated actions or behaviors that have been formed over time, and also include thinking and emotional content, which are stable and not easily changed. Habits can change in response to external stimuli. Some studies have suggested that consumption preference refers to the fact that consumers have a special trust in a particular commodity, store, or trademark, and repeatedly and habitually go to a certain store or repeatedly and habitually buy goods of the same trademark or brand [43]. We used four variables: purchase habits, purchase membership, the share of household vegetable consumption, and the share of low-carbon vegetable consumption as explanatory variables.

#### 2.1.6. Individual and Family Characteristics

Individual and family characteristics are widely considered and included in model analyses in empirical studies of consumer behavior. On the one hand, they can provide a better explanation of consumer behavior [44] and, on the other hand, they can be used as control variables to minimize bias in model estimation [45]. Therefore, we chose individual consumer characteristics (e.g., gender, education, income, etc.,) and household characteristics (household size, etc.,) as explanatory variables for individual perceptions.

### 2.2. Research Scheme Design

Consumer preferences include displayed preferences, which can be obtained by direct observation of consumers’ purchasing behavior, and stated preferences, which can only be obtained by consumers expressing their own intentions. In this paper, a hypothetical value assessment method (CVM) is used to study consumers’ willingness to pay (WTP) for low-carbon leafy greens. Considering that many consumers do not have a high level of awareness of low-carbon vegetables, respondents were first given a message reinforcement and a contextual description. At the place where you often buy vegetables, both “regular leafy greens” and “low-carbon leafy greens” are sold, and the varieties and appearance of these two types of vegetables look the same. The difference is that low-carbon leafy greens have lower carbon emissions (i.e., emissions of greenhouse gases such as nitrous oxide, carbon dioxide, and methane) than conventional leafy greens, with little or no pesticides, fertilizers used in the growing process, manure, and organic fertilizers applied instead, and green pest control techniques used. In the process of storage and transportation, we use cold chain transportation with the lowest possible carbon emissions, and in the process of marketing, we reduce energy and material consumption by reducing the use of packaging.

In this paper, the dichotomous choice method is chosen to guide consumers’ willingness to pay for low-carbon leafy greens. The dichotomous choice method is widely used because respondents’ yes or no responses better simulate market pricing behavior than their direct statement of maximum willingness to pay. The dichotomous choice method simply requires respondents to give a “yes” or “no” answer for a commodity at a different bid price, i.e., by asking respondents “Would you be willing to pay an extra X RMB/kg for low-carbon leafy greens compared to regular leafy greens?/kg?” The question was asked “Would you be willing to pay an extra X RMB/kg for low-carbon leafy greens?” Different bidding prices were given to different samples (RMB 0.5/kg, RMB 1/kg, RMB 2/kg, RMB 4/kg, and RMB 6/kg) in order to examine the decreasing proportion of “yes” responses as the bidding price increased. Out of the 532 valid questionnaires, 105 questionnaires were submitted at RMB 0.50, 105 at RMB 1105, RMB 2105, and RMB 4, and 112 at RMB 6.

### 2.3. Model Construction

Consumers’ willingness to pay for low-carbon leafy greens is a classic dichotomous choice between “willing” and “unwilling”. The maximization of utility is the criterion by which consumers make their purchasing decisions. If consumers choose to buy low-carbon leafy greens in a market where both regular leafy greens and low-carbon leafy greens are available, this implies that low-carbon leafy greens provide greater utility to consumers than regular leafy greens. Accordingly, the following binary logit model was constructed and estimated using Stata 13.0.
(1)$$\ln\left[\frac{P\left(Y=1\right)}{1-P\left(Y=1\right)}\right]=\;a\;+\;bZ+\;cTP+\varepsilon$$

## 3. Survey and Data

### 3.1. Data Collection

The data used in this study came from a socio-economic survey of consumers in Shanghai conducted by the research team from May to September 2017. To ensure the authenticity of the data, survey respondents were selected using a random sample and the survey was conducted in accordance with a face-to-face interview format. Staff training and pre-survey were conducted prior to the formal research. The survey area was the 12 urban areas of Shanghai, including Baoshan, Fengxian, Hongkou, Huangpu, Jiading, Jing’an, Minhang, Pudong, Putuo, Xuhui, Yangpu, and Changning (Figure 2). During the survey, one surveyor took 0.5–1 h to complete a questionnaire, allowing consumers sufficient time to answer information about their consumption choices, with each surveyor completing 5–10 questionnaires per day. A total of 550 questionnaires were distributed for the survey, resulting in 532 valid questionnaires. The survey questions and associated definitions we used are shown in Table 1.

### 3.2. Descriptive Analysis of Consumers’ Perceptions of Low-Carbon Vegetables

The survey found that 61.28% of the consumers surveyed knew or had heard of the concepts of “low-carbon”, “carbon emissions”, and “carbon footprint”. This shows that consumers’ overall awareness of low-carbon concepts is relatively high, which is closely related to the current national efforts to promote ecological civilization and develop ecological agriculture. The survey results (Table 2) show that most consumers believe that low-carbon vegetables are more beneficial to the ecological environment than conventional vegetables, i.e., 21.43% and 51.88% of the respondents “strongly agree” and “somewhat agree” with the statement that “buying low-carbon vegetables is more beneficial to the ecological environment than conventional vegetables”. The majority of consumers believed that the quality and safety of low-carbon vegetables would be significantly higher than that of conventional vegetables, i.e., 19.17% and 40.60% of the respondents “strongly agreed” that “buying low-carbon vegetables would be more secure in terms of quality and safety than conventional vegetables”. In addition, most consumers believe that the price of low-carbon vegetables is significantly higher than conventional vegetables, that purchasing low-carbon vegetables are a consumer trend, and that purchasing low-carbon vegetables is a response to the government’s call for low-carbon vegetables. In addition, most consumers believe that the price of low-carbon vegetables is significantly higher than conventional vegetables. The survey also found that 15.23% of respondents “strongly agreed”, 40.41% “somewhat agreed”, and 33.08% “generally agreed” when asked their opinion on whether they would buy low-carbon vegetables.

## 4. Research Results

### 4.1. Robustness Test of the Model

Before regression estimation, correlation analysis is required in this paper to analyze the possible multicollinearity of each control variable included in the model and to study whether there is a correlation problem. In this paper, the variance inflation factor (VIF) and tolerance (TOL) were used for the correlation test. Table 3 showed that the maximum VIF value was 2.54 and the average VIF value was 1.53, much less than 10. Therefore, there were no serious multicollinearity and correlation problems between variables. The model is within the acceptable range.

### 4.2. Factors Influencing Consumers’ WTP for Low-Carbon Leafy Greens

The model was estimated in this study using Stata 3.0 and the estimation results are shown in Table 4. The pseudo-R2 and LR likelihood values of the model and its *p*-value show that the model has a good fit and overall significance of the variables.

From the model estimation results, it can be seen that eight variables, including bid price (Bpr), vegetable purchase confidence (Buy), ecological protection (Eco), behavioral control (Act), local vegetable purchase habits (Hab), age (Age), and purchase membership (Mem) significantly affect consumers’ willingness to pay for low-carbon leafy greens. Specifically, bid price inversely and significantly affects consumers’ willingness to pay for low-carbon leafy greens; i.e., as the bid price continues to increase, the likelihood that consumers are willing to purchase low-carbon leafy greens continues to decrease in terms of marginal effects. With each increase in bid price (Bpr) by one level, the likelihood that consumers are willing to purchase low-carbon leafy greens decreases by 0.1912 on average. Secondly, consumers who are less confident in the quality and safety of the vegetables they purchase are more willing to pay extra for low-carbon leafy greens, which to some extent also reflects the higher level of confidence in the quality and safety of low-carbon leafy greens. In terms of marginal effects, the likelihood that consumers are willing to pay extra for low-carbon leafy greens increases by an average of 0.1356 for every level of decrease in the level of confidence in the quality and safety of the vegetables. Thirdly, consumers who believe that low-carbon vegetables are better for the environment than conventional vegetables are more likely to be willing to pay extra for low-carbon leafy greens, and, in terms of marginal effects, for every level of increase in consumers’ agreement that “buying low-carbon vegetables is better for the environment than conventional vegetables”, consumers are more likely to be willing to pay extra for low-carbon vegetables. Fourthly, consumers who are more willing to pay extra for low-carbon vegetables are more likely to be willing to pay extra for low-carbon leafy greens. Fifthly, consumers who usually deliberately choose to buy vegetables produced in Shanghai are more likely to be willing to pay extra for low-carbon leafy greens and, in terms of marginal effects, consumers who deliberately choose to buy vegetables produced in Shanghai are on average 0.1673 more likely to be willing to pay extra for low-carbon greens than those who do not deliberately choose to buy vegetables produced in Shanghai. Sixthly, consumers who are the household’s primary vegetable buyers are more likely to be willing to pay extra for low-carbon leafy greens and, in terms of marginal effects, consumers who are the household’s primary vegetable buyers are on average 0.2425 more likely to be willing to pay extra for low-carbon leafy greens than those who are the household’s secondary vegetable buyers. Finally, older consumers are more likely to be willing to pay an additional price for low-carbon leafy greens and, in terms of marginal effects, the likelihood of consumers being willing to pay an additional price for low-carbon leafy greens decreases by an average of 0.051 for each 10-year increase in age.

### 4.3. Group Differences in Consumers’ Willingness to Pay for Low-Carbon Leafy Greens

Based on the average willingness-to-pay formula, this study calculates the average willingness to pay for low-carbon leafy greens among consumers. It can be seen that consumers are willing to pay an additional RMB 2.5367/kg for low-carbon leafy greens compared to conventional leafy greens. Shanghai is currently positioning itself for the development of modern urban agriculture and ecological agriculture is an important tool and the main direction to achieve agricultural modernization. The results of this study show that consumers are aware of low-carbon vegetables and are willing to pay extra for leafy greens that are safer and better for the environment, indicating that there is a market demand for the development of eco-agriculture and the production of low-carbon agricultural products in Shanghai.

In addition to calculating the average willingness to pay for low-carbon leafy greens for all consumers, we also focused on and calculated group differences in willingness to pay for low-carbon leafy greens across consumer groups, including income level, environmental awareness, low-carbon perceptions, purchase of members, vegetable preferences, and individual characteristics, as detailed in Table 5. We choose the above indicators based on the actual survey. Through communication with consumers, we find that these variables are the variables that consumers pay more attention to. Although the effects of certain variables were not significant, willingness to pay for low-carbon leafy greens may still show large differences. The results show that, firstly, the average WTP for low-carbon leafy greens varied significantly between consumers with different environmental awareness and between those who were the primary and secondary buyers of vegetables. Specifically, the group of environmentally conscious consumers was willing to pay an additional RMB 2.6914/kg for low-carbon leafy greens, and the group of non-environmentally conscious consumers was only willing to pay an additional RMB 1.291/kg. The difference between the two was RMB 1.4004. In addition, the group of consumers who were the primary buyers of vegetables were willing to pay an additional RMB 2.8617 for low-carbon leafy greens, while the group of consumers who were the secondary buyers of vegetables were only willing to pay an additional RMB 1.6773/kg for low-carbon leafy greens, a difference of RMB 1.1844 between the two. Secondly, while the average monthly personal income variable did not significantly affect consumers’ willingness to pay for low-carbon leafy greens, the average WTP for low-carbon leafy greens was RMB 0.3079 higher for consumers with incomes of RMB 5000 and above than for those below RMB 5000. The average WTP of low-carbon leafy greens for consumers who also knew about “low-carbon”, “carbon emissions”, and “carbon footprint” was RMB 0.3271 higher than that of consumers who did not know. In addition, male consumers and older consumers had a higher average WTP than female and middle-aged consumers. Finally, the mean WTP for low-carbon leafy greens did not differ significantly between consumers with different vegetable preferences and between consumers of different origins.

## 5. Discussion

The aim of this study was to examine the willingness to pay for low-carbon agricultural products, particularly low-carbon vegetable products, and the factors that influence it among Chinese urban consumers. Our study highlights the fact that, as incomes increase and the level of economic development rises, urban consumers’ overall awareness of the concept of low-carbon products is relatively high, with 61.28% of consumers indicating that they know or have heard of the concepts of “low-carbon”, “carbon emissions”, “carbon footprint”, etc. After reinforcing the concept of low-carbon vegetables, the majority of consumers agreed that low-carbon vegetables are better for ecological protection and quality and safety are better than conventional vegetables. A total of 21.43% and 51.88% of the respondents “strongly agreed” and “somewhat agreed” to the statement that “buying low-carbon vegetables is better for ecological protection than conventional vegetables”. A total of 19.17% and 40.60% of the respondents “strongly agreed” and “somewhat agreed” with the statement that “Buying low-carbon vegetables is more secure in terms of quality and safety than buying conventional vegetables”. This indicates that the majority of consumers purchase low-carbon vegetables for ecological reasons. These findings have been confirmed in studies in other countries [2,4,5].

However, bid prices, subjective norms, and consumer preferences also significantly influence consumer purchasing behavior. For example, variables such as consumer trends and purchasing habits of locally produced vegetables significantly influence consumers’ willingness to pay for low-carbon leafy greens. Specifically, as bid prices continue to increase, the likelihood of consumers being willing to purchase low-carbon leafy greens decreases. In addition, respondents were more likely to choose to buy locally produced vegetables in Shanghai due to their diet of leafy greens and their “ease of access”. Older people are more likely to focus on “eco-friendly” factors than younger people [15,28], and this study suggests that older consumers are more likely to be willing to pay extra for low-carbon leafy greens. Consumers who are the household’s primary vegetable buyers are also more willing to pay extra for low-carbon leafy greens due to health and nutrition concerns. This fact translates into a growing consumer interest in environmentally friendly or low-carbon products, a phenomenon also observed by Eftimov et al. [45].

Even though the eco-consciousness variable did not significantly affect consumers’ willingness to pay for low-carbon leafy greens, the potential and indirect impact of eco-consciousness on consumers’ willingness to pay for low-carbon leafy greens cannot be ignored. By dividing consumers into two groups, the strongly and the weakly environmentally conscious, and comparing group differences, we found that the ecological conservation variable significantly influenced the willingness to pay for low-carbon leafy greens for the strongly environmentally conscious and, conversely, the variable was not significant for the weakly environmentally conscious group. In addition, the calculation of the average willingness to pay yielded that consumers were willing to pay an additional RMB 2.5367/kg for low-carbon leafy greens compared to conventional leafy greens. This is within an acceptable price range for the average Chinese household. Interestingly, the average willingness to pay RMB 1.1844/kg for low-carbon leafy greens is higher for the group of consumers who are the household’s primary buyers of vegetables than for the group of consumers who are the household’s secondary buyers of vegetables. This shows that consumers are not only concerned about the environmental benefits of low-carbon vegetables, but also concerned about the nutritional and health benefits of low-carbon vegetables. This view is also reflected in previous studies in the literature [38,40,42].

## 6. Conclusions

It is becoming increasingly clear that the development of a low energy consumption, low emission, and low pollution economic model has far-reaching implications for the ecological and socio-economic development of the country. The consumer, as the ultimate demander of low-carbon agricultural products, determines the existence and long-term development of the market for low-carbon agricultural products. Whether for economic or environmental reasons, low-carbon agricultural products, such as green and organic products, will become more sustainable consumer products.

Firstly, as an international metropolis, Shanghai, China should set an example and lead the way in the development of modern urban green agriculture, grasp the trend of ecological consumption, vigorously develop low-carbon agriculture, attach importance to the production of low-carbon vegetables, give policy support to business entities that actively explore and produce low-carbon vegetables, and encourage support for cooperation between industry, academia, and research to research and promote good models of low-carbon vegetable production.

Secondly, the government should step up its efforts to promote and guide the production of low-carbon vegetables and adopt different strategies for different consumer groups in an effort to raise consumers’ awareness of environmental protection and their willingness to pay for low-carbon vegetables. The awareness of low-carbon vegetables among Shanghai residents has yet to be raised, which requires the government to make full use of channels such as the internet, television, newspapers, and magazines to increase publicity and raise consumer demand for low-carbon vegetables and even low-carbon agricultural products, so as to create favorable market conditions for the development of a low-carbon vegetable industry. At the same time, in view of the differences in willingness to pay for low-carbon vegetables among consumer groups with different income levels, environmental awareness, low-carbon knowledge, vegetable preferences, and individual characteristics, there is a need to focus and target publicity and guidance, especially for consumer groups with low income, low environmental awareness, the household’s secondary vegetable purchasers, women, young and middle-aged people, and those with high school education or below, to promote low-carbon vegetables to them in terms of protection of the ecological environment and the improvement of quality and safety, and increase their level of willingness to pay for low-carbon vegetables. In addition, efforts are made to raise consumers’ awareness of environmental protection, which plays an important role in increasing the overall level of consumers’ willingness to pay for low-carbon vegetables.

Finally, quality and safety supervision is being strengthened to improve the quality of vegetables in China. However, the survey found that Shanghai residents do not rate the quality and safety of vegetables in the market very highly, and are not entirely confident about the quality and safety of the vegetables they buy. The use of chemical fertilizers and pesticides by vegetable producers is still relatively common, both at the national level and in Shanghai, which poses certain hidden risks to vegetable quality and safety, and hinders the development of the low-carbon vegetable industry. We should continue to strengthen efforts to test and monitor the quality and safety of vegetables on the market, and continue to increase the proportion of vegetables tested and the types of pesticides tested.

## Figures and Tables

**Figure 1 ijerph-20-00358-f001:**
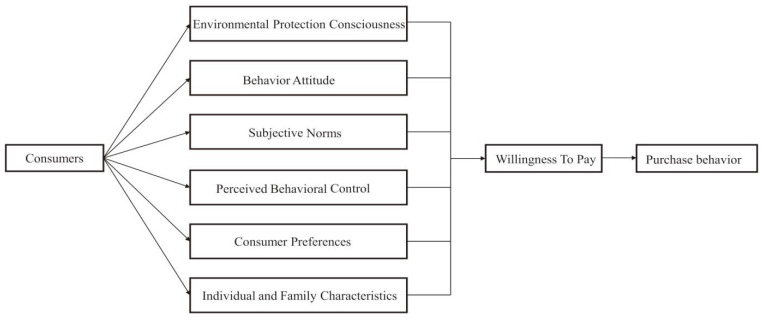
An analytical framework for influencing factors of consumers’ willingness to pay.

**Figure 2 ijerph-20-00358-f002:**
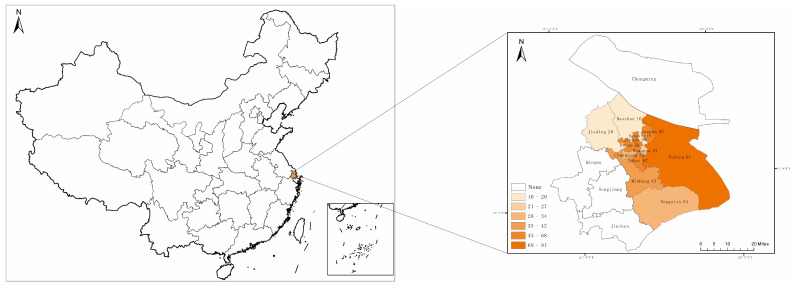
Map of sample area and sample size distribution.

**Table 1 ijerph-20-00358-t001:** Definition of relevant indicators.

Item	Subitem	Definition	Assignment	Abbreviation
Bid Price		RMB 0.5, 1, 2, 4, and 6 (Unit: RMB/kg)	Actual Values	Bpr
Environmental Protection Consciousness	Environmental awareness	I have strong environmental awareness: ① strongly disagree ② not really agree ③ generally agree ④ somewhat agree ⑤ strongly agree	Assign a value of 1 to 5	Epc
	Buy with confidence	Are you confident about the quality and safety of the vegetables you buy? ① very confident ② quite confident ③ fairly confident ④ not very confident ⑤ very unsure	Assign a value of 1 to 5	Buy
	Low-carbon awareness	Do you know or have you heard of the concepts of “low-carbon” “carbon emissions” “carbon footprint” etc.? ① know (heard of it) ② do not know (not heard of it)	Know = 1, Do not know = 0	Loc
Behavior Attitude	Ecological protection of the environment	Buying low-carbon vegetables is better for ecological conservation than buying conventional vegetables: ① strongly disagree ② not really agree ③ generally agree ④ somewhat agree ⑤ strongly agree	Assign a value of 1 to 5	Eco
	Quality and safety assurance	Buying low-carbon vegetables is more secure in terms of quality and safety than buying conventional vegetables: ① strongly disagree ② not really agree ③ generally agree ④ somewhat agree ⑤ strongly agree	Assign a value of 1 to 5	Qua
Subjective Norms	Consumer trends	Buying low-carbon vegetables is a consumer trend: ①strongly disagree ②not really agree ③generally agree ④somewhat agree ⑤ strongly agree	Assign a value of 1 to 5	Tre
	Government call	Buying low-carbon vegetables is a response to the government’s call: ① strongly disagree ② not really agree ③ generally agree ④ somewhat agree ⑤ strongly agree	Assign a value of 1 to 5	Gov
Perceived Behavioral Control	Price expectations	Buying low-carbon vegetables is significantly more expensive than buying conventional vegetables: ① strongly disagree ② not really agree ③ generally agree ④ somewhat agree ⑤ strongly agree	Assign a value of 1 to 5	Exp
	Behavioral control	If low-carbon vegetables were available in the market, I would buy them decisively: ① strongly disagree ② not really agree ③ generally agree ④ somewhat agree ⑤ strongly agree	Assign a value of 1 to 5	Act
Consumer Preferences	Local food buying habits	Do you usually choose to buy vegetables produced in Shanghai on purpose? ① Yes ② No	Will = 1, Will not = 0	Hab
	Purchase of members	Are you the main person in your household who buys vegetables? ① Yes ② No	Yes = 1, No = 0	Mem
	Share of vegetable consumption	Share of vegetable consumption in your household food expenditure.	50% and above = 1, other = 0	Pro
	Share of leafy greens consumption	Consumption of leafy greens as a proportion of your total household expenditure on vegetables.	50% and above = 1, other = 0	Gre
Individual and Family Characteristics	Gender	Gender: ① Male ② Female	Male = 1, Female = 0	Sex
	Age	Age: Unit weeks	Actual values	Age
	Education	Education: ① Elementary school and below ② Junior high school ③ Secondary/high school ④ Junior college ⑤ Bachelor’s degree ⑥ Graduate	Assign a value of 1 to 6	Edu
	Income level	Average monthly personal income (after tax): RMB in units	Actual values	Inc
	Place of origin	Place of origin: ① local (Shanghai) ② non-local	Shanghai local = 1, other = 0	Pla
	Number of family members	Total household size (living together)	Actual values	Fam
	Kid situation	Are there any children in the household (15 years old and below): ① Yes ② No	Yes = 1, No = 0	Kid
	Situation of the elderly	Is there an elderly person in the household (aged 60 and above, referring to elders): ① Yes ② No	Yes = 1, No = 0	Old

**Table 2 ijerph-20-00358-t002:** Consumer perceptions of low-carbon vegetables.

Title	Buying Low-Carbon Vegetables Is More Eco-Friendly Than Buying Conventional Vegetables		Buying Low-Carbon Vegetables Will Provide Greater Assurance of Quality and Safety Than Buying Conventional Vegetables		Buying Low-Carbon Vegetables Is Significantly More Expensive Than Buying Conventional Vegetables		Buying Low-Carbon Vegetables Is a Consumer Trend		Buying Low-Carbon Vegetables Is a Response to the Government’s Call	
	Frequency	Proportion	Frequency	Proportion	Frequency	Proportion	Frequency	Proportion	Frequency	Proportion
		(%)		(%)		(%)		(%)		(%)
Strongly disagree	15	2.82	24	4.51	6	1.13	15	2.82	17	3.2
Not really agree	40	7.52	31	5.83	61	11.47	47	8.83	76	14.29
Generally agree	87	16.35	159	29.89	109	20.49	170	31.95	149	28.01
Somewhat agree	276	51.88	216	40.6	253	47.56	209	39.29	217	40.79
Strongly agree	114	21.43	102	19.17	103	19.36	91	17.11	73	13.72

**Table 3 ijerph-20-00358-t003:** Measurement results of the VIF and TOL for each variable.

Variable	VIF	1/VIF
Epc	1.92	0.52
Buy	1.07	0.94
Loc	1.12	0.89
Eco	1.97	0.51
Qua	2.54	0.39
Tre	1.53	0.65
Gov	1.66	0.60
Exp	1.82	0.55
Act	2.00	0.50
Hab	1.16	0.86
Mem	1.19	0.84
Pro	1.11	0.90
Gre	1.07	0.93
Sex	1.07	0.94
Age	1.86	0.54
Edu	1.31	0.77
Inc	1.27	0.79
Pla	1.15	0.87
Fam	1.82	0.55
Kid	1.69	0.59
Old	1.52	0.66
Mean VIF	1.53	

**Table 4 ijerph-20-00358-t004:** Regression results of influencing factors.

Variables	Coefficient	Z-Value	*p*-Value	Marginal Probability
**Bpr**	−0.7698 ***	−9.56	0	−0.1912
**Epc**	−0.1415	−0.79	0.43	−0.0352
**Buy**	0.5458 ***	3.91	0	0.1356
**Loc**	−0.0296	−0.12	0.908	−0.0074
**Eco**	0.4149 **	2.44	0.015	0.1031
**Qua**	−0.0656	−0.37	0.713	−0.0163
**Tre**	−0.2087	−1.44	0.151	−0.0518
**Gov**	−0.0968	−0.64	0.523	−0.0241
**Exp**	−0.1466	−0.88	0.377	−0.0364
**Act**	0.5852 ***	3.51	0	0.1454
**Hab**	0.6766 ***	2.67	0.008	0.1673
**Mem**	1.0301 ***	3.46	0.001	0.2425
**Pro**	−0.2882	−1.01	0.311	−0.0709
**Gre**	−0.1828	−0.75	0.454	−0.0454
**Sex**	0.4054	1.63	0.102	0.1004
**Age**	−0.0206 *	−1.87	0.061	−0.0051
**Edu**	0.1508	1.61	0.107	0.0375
**Inc**	0.000003	−0.43	0.667	0.000001
**Pla**	0.1698	0.66	0.508	0.0421
**Fam**	−0.1211	−1.09	0.275	−0.0301
**Kid**	−0.1195	−0.42	0.673	−0.0297
**Old**	0.0318	0.11	0.916	0.0079
**Constant term**	−0.9619	−0.87	0.383	
**Pseudo R2**	0.3551			
**LR chi2**	261.78			
**Prob > chi2**	0			

Note: *, ** and *** denote 10%, 5%, and 1% significance levels, respectively.

**Table 5 ijerph-20-00358-t005:** Group differences in consumers’ average willingness to pay under different variable categories.

Influencing Factors	Variable Category	Frequency	Proportion	Willingness to Pay Level (RMB/kg)
Income level (Inc)	Under RMB 5000	249	46.80%	2.3437
	RMB 5000 and above	283	53.20%	2.6516
Environmental awareness (Epc)	Not environmentally conscious	111	20.86%	1.291
	Environmentally conscious	421	79.14%	2.6914
Low-carbon awareness (Loc)	Do not know	206	38.72%	2.2566
	Know	326	61.28%	2.5837
Primary buyer (Mem)	Not a primary buyer of the family	138	25.94%	1.6773
	Is the primary buyer of the family	394	74.06%	2.8617
Share of vegetable consumption (Pro)	Vegetable consumption <50% of household food expenditure	419	78.76%	2.5931
	Vegetable consumption ≥50% of household food expenditure	113	21.24%	2.44
Share of consumption of leafy greens (Gre)	Consumption of leafy green vegetables <50% of household vegetable expenditure	219	41.17%	2.4939
	Consumption of leafy green vegetables ≥50% of household vegetable expenditure	313	58.83%	2.5586
Gender (Sex)	Female	276	51.88%	2.338
	Male	256	48.12%	2.6343
Situation of the elderly (Old)	Young and middle-aged (under 60 years old)	432	81.20%	2.4851
	Elderly (60 years and above)	100	18.80%	2.8436
Education (Edu)	High school and above	247	46.43%	2.2768
	Tertiary/undergraduate and above	285	53.57%	2.7079
Place of origin (Pla)	Local (Shanghai)	192	36.09%	2.5517
	Non-local	340	63.91%	2.5085

## Data Availability

The data presented in this study are available on request from the corresponding author. The data are not publicly available as “the rest of the team also needs to write papers with this data.

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
