# Peer review of "Influencing Factors and Group Differences of Urban Consumers’ Willingness to Pay for Low-Carbon Agricultural Products in China"

_ijerph, 2022, doi:10.3390/ijerph20010358_

Round 1

Reviewer 1 Report

The authors investigate the factors that influence Chinese urban consumers' willingness to pay for low-carbon products. Specifically, the findings show that urban consumers' awareness of low-carbon products, preferences, subjective norms, and bid prices have a significant impact on consumers' willingness to pay. The authors present a survey study to support their theorizing and hypotheses. The topic is both theoretical and practical importance. But, I do have some concerns, roughly divided into conceptual and empirical.

Conceptual issues

1. Although the ideas are interesting, the paper in its current form is not making as much of a theoretical contribution as it potentially could. In particular, the authors review the literature on consumers' willingness to pay for low-carbon products (lines 114-133). Previous research has demonstrated that several factors (e.g., carbon labels, psychological attribution, policy interventions, awareness, consumer preferences, education, behavioral intention, age,  gender, etc.) influence consumers' willingness to pay for low-carbon products. Given that some factors identified by the current research (e.g., consumer preferences, gender, education, and awareness) are remarkably similar to those found by the existing literature, the authors should better clarify the theoretical contribution of the current manuscript.

2. The definition of some constructs in the current research are vague. For example. in "2.1.5. Consumer Preferences," the authors investigate behavioral habits instead. What is the relationship between consumer preferences and behavioral habits? The authors should unify the constructs used. Furthermore, the current research includes individual consumer characteristics (e.g. gender, education, income, etc.) and household characteristics (household size, etc.) in the construct of individual cognition. However,  individual consumer characteristics and household characteristics are demographic variables (Smelser & Baltes 2001), which does not belong to consumer cognition.

3. The authors mention green products (lines 61-66). At first glance, green products are similar to low-carbon products. Even though the current paper defines low-carbon products, it fails to clarify the definition differences between green products and low-carbon products. I suggest the authors remove content regarding green products if they are not theoretically related to low-carbon products.

4. One of the research focuses of this paper is the low-carbon products. In the theoretical model (Figure 1), however, the constructs are not theoretically related to low-carbon products. Moreover, the research model does not show any characteristics of low-carbon products. In a nutshell, the research model can be apply to any consumption context. The authors should further explain why the research model is theoretically important to the research on low-carbon products but not other products.

5. The dependent variable "purchase behavior" is not accurate as the method of the research is survey. Survey can only measure purchase tendency but not behavior.

Empirical issues

1. How the measures of each construct are developed? Are they adopted or adapted from previous research? Or are they created by the authors? The authors should explain the process of how they select questionnaire items.

2. As some constructs (e.g., perceived behavioral control) are measured by more than two items, the authors should provide the results of reliability and validity tests to make sure the measures meet the requirements of reliability and validity thresholds.

3. A one-sided survey can cause common method bias. I suggest the authors use the marker variable method to assess the common method bias (Lindell & Whitney 2001).

4. Why do they authors choose urban Chinese consumers? Can the results of this research generalize to rural Chinese consumers? This question is both theoretically and empirically important.

5. The authors should add description information of the sample (e.g., age, education, gender, income, occupation, etc.).

Minor issues

1. There are several typos in the manuscript. For example, there are two commas after "use of carbon label" (line 71) and there is a full stop after "Creating value" (line 120).

2. I highly suggest the authors to find a copy editor to assist you with any grammar and language issues.

References

Lindell, M. K., & Whitney, D. J. (2001). Accounting for common method variance in cross-sectional research designs. Journal of Applied Psychology86(1), 114-121.

Smelser, N. J., & Baltes, P. B. (Eds.). (2001). International encyclopedia of the social & behavioral sciences (Vol. 11). Amsterdam: Elsevier.

Author Response

Dear reviewer,

Best,

Ning Geng

Reviewer 2 Report

I found the manuscript interesting for the topic dealt with. The paper present useful information on the influence of environmental awareness and consumption preferences in food choices.

In particular:

Introduction contextualizes well the subject and gives appropriate starting scientific basis; the section provides sufficient background and includes all relevant references. Beside this, only check ref. n. 5 and 6 because in my opinion seem not related to what is written in the text.

Materials and methods are well structured and adequately described.

I suggest to specify the variables more clearly in paragraph 2.1.2 as you did in the other paragraphs.

Results are presented in a averagely clear way.

In order to facilitate the reading, I suggest including an overall table of all the considered variables.

Moreover, correct the sentence from lines 341 to 345, because the variable "Tre" is not significant.

Discussion effectively contextualizes the results obtained and strive to compare the results with other studies, perhaps somewhat concisely.

Conclusions are adequately supported by the results.

Margin comment: correct punctuation extensively (many missing points)

Author Response

Dear reviewer,

Best,

Ning Geng

Round 2

Reviewer 1 Report

After reading your manuscript, I found you have well addressed the issues raised in the first round. Please do a final copy-editing and particularly double-check the references to ensure they are all correct.